Genome‑wide analysis of the MYB gene family in pumpkin

Xu Minyan 1 2
Fu Jingjing 1
Ni Ying 2
Zhang Chenchen 1 zhangcc2015@163.com
1 Laboratory of Botany, Anhui Wenda University of Information Engineering , Hefei, Anhui , China
2 School of Life Sciences, Anhui Agricultural University , Hefei, Anhui , China
Balao Francisco
Electronic publication date: 2024 Apr 25
Publication date: 2024
Volume: 12
Electronic Location ID: e17304
Received 2023 Aug 25; Accepted 2024 Apr 4
Copyright: © 2024 Xu et al.
Copyright year: 2024
Copyright holder: Xu et al.
License: This is an open access article distributed under the terms of the Creative Commons Attribution License, which permits unrestricted use, distribution, reproduction and adaptation in any medium and for any purpose provided that it is properly attributed. For attribution, the original author(s), title, publication source (PeerJ) and either DOI or URL of the article must be cited.
License URL: https://creativecommons.org/licenses/by/4.0/

Keywords: MYB, Gene, Cis-acting elements, ABA, SA, MeJA, Abiotic/biotic stresses, Transcription factors, Genome-wide analysis, Pumpkin

Funding: 2020 Outstanding Talent Support Program for Universities gxyq2020086 This work was supported by the 2020 Outstanding Talent Support Program for Universities (No. gxyq2020086). The funders had no role in study design, data collection and analysis, decision to publish, or preparation of the manuscript.

==============================
The MYB gene family exerts significant influence over various biological processes and stress responses in plants. Despite this, a comprehensive analysis of this gene family in pumpkin remains absent. In this study, the MYB genes of Cucurbita moschata were identified and clustered into 33 groups (C1-33), with members of each group being highly conserved in terms of their motif composition. Furthermore, the distribution of 175 CmoMYB genes across all 20 chromosomes was found to be non-uniform. Examination of the promoter regions of these genes revealed the presence of cis-acting elements associated with phytohormone responses and abiotic/biotic stress. Utilizing quantitative real-time polymerase chain reaction (qRT-PCR), the expression patterns of 13 selected CmoMYB genes were validated, particularly in response to exogenous phytohormone exposure and various abiotic stressors, including ABA, SA, MeJA, and drought treatments. Expression analysis in different tissues showed that CmoMYB genes are expressed at different levels in different tissues, suggesting that they are functionally divergent in regulating growth and abiotic stresses. These results provide a basis for future studies to characterize the function of the MYB gene family under abiotic stresses in pumpkins.

Introduction

Transcription factors (TFs) are characterized by the presence of distinct structural domains, including a nuclear localization signal domain, DNA-binding domain, transcription regulation domain, and oligomerization domain (Ptashne, 1988; Du et al., 2012). Depending on the structure of the domain that specifically binds to DNA sequences, the TFs can be classified into many families (Liu, White & MacRae, 1999). Among these families, the MYB superfamily stands out, boasting conserved MYB DNA-binding structural domains that are ubiquitously distributed across eukaryotes (Riechmann et al., 2000; Dubos et al., 2010). The identification of the MYB gene originated from the avian myeloblastosis virus (Klempnauer, Gonda & Michael Bishop, 1982), followed by the discovery of A-MYB, B-MYB, and c-MYB, in numerous vertebrates, showcasing their involvement in the regulation of cell differentiation, proliferation, and apoptosis (Golay et al., 1991; Tanaka et al., 1999; Rushton et al., 2003; Davidson et al., 2005). The first functional MYB gene found in plants was ZmMYBC1 that involved in the regulation of anthocyanin biosynthesis (Paz-Ares et al., 1987). The MYB domain typically comprises one to four imperfect repeats, each forming a helix-turn-helix structure spanning 50–53 amino acids (Stracke et al., 2014; Ogata et al., 1996). Notably, these repeats feature uniformly localized tryptophan residues, which congregate within the hydrophobic core of each repeat, thereby stabilizing the three-dimensional helix-turn-helix structure of the MYB protein’s DNA binding domain. With the proliferation of genome sequencing endeavors, MYB family TFs have been extensively investigated across numerous plant species. Notable examples include Arabidopsis thaliana (193 AtMYBs) (Chen et al., 2006), Oryza sativa (197 OsMYBs) (Katiyar et al., 2012), Glycine max (252 GmMYBs) (Du et al., 2012), Gossypium hirsutum (524 GhMYBs) (Salih et al., 2016), Solanum tuberosum (253 StMYBs) (Li et al., 2020), Pyrus bretschneideri (129 PbMYBs) (Cao et al., 2016).

Previous studies have established the pivotal role of the MYB gene family in cell development, primary and secondary metabolism, and stress responses (Dubos et al., 2010; Mao et al., 2011; Yang, Dai & Zhang, 2012). Notably, the overexpression of SpMYB (from Solanum pimpinellifolium L3708) demonstrated heightened resistance to necrotrophic pathogens and enhanced salt and drought stress tolerance in tobacco (Liu et al., 2016; Li, Luan & Yin, 2014). Similarly, transgenic rice plants overexpressing OsMYB2P-1 exhibited increased tolerance to inorganic phosphate starvation through the promotion of primary root elongation (Dai et al., 2012). In Arabidopsis, AtMYB60 and AtMYB61 have been implicated in root development and stomatal aperture (Liang et al., 2005; Simeoni et al., 2022), while AtMYB44, AtMYB52, and AtMYB96 have been associated with drought stress response via ABA-mediated pathways (Joo et al., 2017; Park, Kang & Kim, 2011; Choudhary & Senthil-Kumar, 2021). Despite extant studies on the MYB gene family, a comprehensive genome-wide characterization of this family remains absent in C. moschata.

Pumpkins, recognized for their escalating economic importance due to bioactive compounds such as carotenoids, phenolic compounds, and flavonoids, also serve as rootstocks for other cucurbit crops, contributing to enhanced tolerance against soil-borne diseases and abiotic stresses (Ninčević Grassino et al., 2023; Guo et al., 2023). In recent years, water scarcity has become increasingly important for crops because of global climate issues. The yield and quality of pumpkin can be severely affected by drought stress, so it is important to identify candidate genes responsible for drought stress tolerance. Highlighting the crucial role of the MYB gene family in stress responses, this research specifically concentrates on the comprehensive genome-wide identification and expression analysis of MYB transcription factors (TFs) in pumpkins under abiotic stress conditions. The investigation encompasses the construction of phylogenetic relationships among CmoMYB proteins, along with an exploration of chromosome localization, genomic structure, and motif protein composition. Subsequently, thirteen selected MYB TFs undergo scrutiny, and their expression patterns under abiotic stress are meticulously analyzed using qRT-PCR. A notable pattern of regulation emerged, with 10 genes exhibiting up-regulation and one gene demonstrating down-regulation following ABA treatment at 6 h. Additionally, 11 CmoMYB genes displayed significant regulation, comprising nine up-regulated and two down-regulated genes, excluding CmoMYB59 (which initiated a decrease at 6 h) and CmoMYB3 (which initiated an increase at 6 h) after jasmoninc acid (JA) treatment. After salicylic acid (SA) treatment, all CmoMYB genes exhibited significant regulation at 3 h, with 11 genes up-regulated and two down-regulated. This study aims to lay the foundation for understanding the regulatory mechanism of the MYB gene.

Materials and Methods

Identification and characteristics analysis of the MYB gene family in C. moschata

The protein sequence database of C. moschata was downloaded from the Cucurbitaceae genome database (http://cucurbitgenomics.org/). In addition, the MYB sequences of Arabidopsis thaliana and Oryza sativa were obtained from the phytozome database (https://phytozome-next.jgi.doe.gov/). The CmoMYB was identified from two plants at the whole genome level using two approaches. First, functionally known MYB protein sequences of Arabidopsis (AtMYB4, AT4G38620) and Oryza sativa (OsMYB, LOC_Os08g43550) were used as query sequences to blast the proteins of the C. moschata. All domains of primary candidate MYB proteins were submitted to the Pfam domain database (http://pfam.xfam.org/) and the Conserved Domains Database (CDD, http://www.ncbi.nlm.nih.gov/Structure) and the profile hidden Markov models (HMMER, https://www.ebi.ac.uk/Tools/hmmer/) to further examined. All MYB protein sequences containing MYB or MYB-like domains were selected and named based on the homology of AtMYB4, respectively.

The ID numbers, genomic positions, genome sequences, and protein sequences of CmoMYB members were downloaded from the Cucurbita genome database. The ExPASy proteomics online tool (https://web.expasy.org/protparam/) was used to analyze the physiological and biochemical characteristics of CmoMYB proteins, including number of amino acids, the relative molecular mass (Mw), coding sequence (CDS), and isoelectric point (pI), etc. The subcellular localization of ComMYB proteins was predicted by using the WoLF PSORT website (https://wolfpsort.hgc.jp/) and BUSCA website (http://busca.biocomp.unibo.it/).

Construction of phylogenetic trees of MYB proteins and duplication analysis

MYB protein phylogenetic trees were constructed and CmoMYB proteins were grouped to explore the evolutionary relationships between CmoMYB and AtMYB proteins. Full-length CmoMYB and AtMYB protein sequences were implemented for multiple sequence alignment using the ClustalW Program in the software MEGA, and the self-help method of phylogenetic experiments (following parameters: p-distance, partial deletion, and bootstrap = 1,000) was used to construct the neighbor-joining (NJ) phylogenetic tree. The tree was visualized and optimized through the ChiPlot (https://www.chiplot.online/#Phylogenetic-Tree).

For the duplication analysis of the pumpkin MYB genome, we employed BLAST to compare the genomic sequences and retrieved essential genomic features such as chromosome length, position, inter-chromosomal linkage, and gene annotations from the GFF3 files. The integration of these files, along with the gene link format file, GFF file format file, and Chr-Layout format file was performed using the File Merge function of the MCScanX package in TBtools (v1.108) to investigate MYB gene duplication events in pumpkin.

Chromosomal location, genetic structure, conserved motifs, and cis-acting regulating element prediction

To investigate the genetic structure and chromosomal localization of the MYB gene, gff3 files about C. moschata were acquired from the Cucurbitaceae database. Subsequently, the conserved motifs within CmoMYB proteins were delineated utilizing the Multiple Em for Motif Elicitation (MEME) Program (https://meme-suite.org/meme/tools/meme), with a specified parameter setting of 10 as the maximum number of conserved motifs to be identified. Furthermore, the 2 kbp promoter sequences associated with CmoMYB genes were obtained. The PlantCARE database website (https://bioinformatics.psb.ugent.be/webtools/plantcare/html/) served as the platform for the prediction of cis-acting elements. TBtools software (v1.108) was used to visualize the above results.

Plant materials, abiotic stress, and hormones treatment

The expression levels of CmoMYBs were investigated under three hormones and abiotic stress, including ABA, SA, JA, and simulated drought. The ‘TianMiyihao’ was used as the experimental material. Seeds were sterilized with 75% ethanol, then rinsed five times with sterile water and germinated on Petri dishes. After 3 days, seedlings with good growth conditions were cultured with Hoagland’s nutrient solution in a greenhouse (28 °C, 70–80% humidity, 16 h light/8 h dark). The roots, stems, and leaves of pumpkin were harvested to analyze the expression levels of CmoMYB genes. Meanwhile, uniformly sized three-leaf seedlings were treated with simulated drought treatment (20% PEG6000) and different hormones (100 µM JA, 10 µM ABA, and 100 µM SA). Leaves from each group were obtained after 0, 3, and 6 h of treatment, respectively. The samples were snap-frozen in liquid nitrogen and stored at −80 °C. Each sample consisted of three different plant leaves and three experiments were performed.

RNA extraction, cDNA synthesis, qRT-PCR, and Expression analysis

Total RNA from each sample was extracted by using Trizol reagent (Takara, Beijing, China), and the cDNA was synthesized by using the First-Strand cDNA Synthesis Kit (Vazyme, Nanjing, China). The specific primers for the ComMYB genes were designed using Primer Premier 5 software (Table S2) for qRT-PCR analysis. The reaction consisted of 10 µL AceQ qPCR SYBR Green Master Mix (Vazyme, Nanjing, China), 0.2 µmol L−1 upstream and downstream primers, 2.0 µL cDNA, and up to 20 µL with ddH2O. The qRT-PCR process was set as follow steps: 95 °C for 3 min, followed by 40 cycles of 95 °C for 10 s, 60 °C for 30 s, and 60 °C for 30 s. The Cmoβ-actin gene was used as the normalization reference gene. The relative expression level of genes was calculated by the 2−∆∆Ct method. Three experimental replicates were performed for each sample. For tissue-specific expression analysis of 175 CmoMYB genes, the expression data for four tissues (leaves, roots, stem, and fruit) were obtained from the Cucurbitaceae database (Sun et al., 2017). The TBtools software was used to construct a heatmap for specific expression analysis.

Results

Identification of the CmoMYB gene family

To identify members of the CmoMYB gene family, we conducted a comprehensive search of the entire set of Cucurbita moschata protein sequences to identify proteins harboring either MYB or MYB-like domain sequences, which served as initial candidate proteins. Subsequently, candidate proteins underwent further validation through the HMMER website, with proteins containing divergent conserved domains being excluded from the analysis. A total of 175 CmoMYB proteins were successfully identified and individually designated. Detailed information regarding the CmoMYB genes is provided in Table S1. As shown in Table S1, a total of 175 CmoMYB proteins were successfully identified (see Table S1 for details), including 1R-MYB (21), R2R3-MYB (149), 3R-MYB (4), and 4R-MYB (1). The lengths of the CmoMYB sequences ranged from 249 to 3,117 base pairs (bp), corresponding to polypeptide lengths spanning from 82 (CmoMYB171, CmoMYB172) to 1,038 (CmoMYB150) amino acids, with an average length of 343 amino acids. Furthermore, the calculated molecular weights (Mw) of the CmoMYB proteins ranged from 9.80 to 114.86 kilodaltons (kDa), while the theoretical isoelectric points (pI) ranged from 4.61 (CmoMYB30) to 11 (CmoMYB163). In addition to being distributed in the nucleus, some proteins were also distributed in the cytoplasm (Table S1).

To explore the orthologous relationship between CmoMYB proteins and their counterparts in Arabidopsis thaliana (AtMYB proteins), we employed the neighbor-joining method to construct a phylogenetic tree (Fig. 1). Analysis of the phylogenetic tree revealed the classification of MYB proteins from both species into 33 distinct categories. Among these, 126 AtMYB proteins were allocated into 25 subgroups. Notably, the majority of CmoMYB proteins (depicted by purple circles in Fig. 1) clustered together with their homologous AtMYB proteins (marked by green circles in Fig. 1), indicating evolutionary conservation. However, exceptions were observed in clades C16 (comprising CmoMYB64, CmoMYB74, and CmoMYB159) and C24 (encompassing CmoMYB4, CmoMYB5, and CmoMYB6), which exclusively contained CmoMYB proteins. This observation suggests that MYB proteins may have undergone distinct evolutionary adaptations to environmental changes.

Figure 1 The Neighbor-Joining phylogenetic tree of MYB proteins between C.moschata and Arabidopsis.

The phylogenetic tree was constructed based on the MYB domain alignment by the MEGA7.0 program with a bootstrap test (replicated 1,000 times). The proteins are clustered into 33 subgroups (e.g., C1), and 25 clades of AtMYBs are labeled in the evolutionary tree (e.g., S1). Filled purple circles represent the MYB proteins of C.moschata, and filled black circles represent the MYB proteins of Arabidopsis.

Chromosomal localization

The 175 CmoMYB genes were mapped to the twenty chromosomes of Cucurbita moschata, with varying gene distributions across each chromosome. The distribution patterns of CmoMYB genes on chromosomes revealed that chromosome 14 harbored the highest number of CmoMYB genes (20), followed by chromosome 1 (17), and the least number of CmoMYB genes were found on chromosome 16 (3). Additionally, there were 14 genes each on chromosomes 2, 4, and 11, respectively; six genes each on chromosomes 5 and 19, respectively; five genes each on chromosomes 3, 7, 8, 9, and 17, respectively; and two genes each on chromosomes 10 and 12, respectively. Furthermore, chromosomes 6, 13, 15, 18, and 20 contained 10, 8, 11, 4, and 9 genes, respectively. Several CmoMYBs are clustered in different specific locations, such as the top of chromosomes 1 and 14, and the bottom of chromosomes 4 and 11. For the selection of duplicated MYB gene pairs, MYB duplicates were scrutinized within the pumpkin genome, resulting in the detection of 135 homologous MYB gene pairs distributed across 20 chromosomes (refer to Fig. S2 and Table S3). Among these, 142 genes exhibited one to four homologous MYB counterparts. Such as CmoMYB8 and CmoMYB50 both have four homologous MYB genes.

Analysis of genetic structure and conserved domains of CmoMYBs

The genetic structure and conserved motif composition of CmoMYB genes were analyzed for 175 CmoMYBs (Fig. 2). Analysis of the structural characteristics of these genes revealed that the vast majority (94%, 165) exhibited intron numbers ranging from 1 to 14, with CmoMYB168 presenting the highest count of both exons (15) and introns (14). Conversely, a subset of 11 CmoMYB genes possessed a solitary exon, comprising CmoMYB56, CmoMYB126, CmoMYB127, CmoMYB131, CmoMYB132, CmoMYB139, CmoMYB144, CmoMYB148, CmoMYB160, CmoMYB165. The majority (85 of the 175 CmoMYBs) of CmoMYBs were typically spliced with three exons and two introns. Moreover, most CmoMYBs (85%) had no more than three introns.

Figure 2 The conserved motifs and exon/intron structures of MYB genes.

(A) The distribution pattern of conserved motifs in the CmoMYBs was identified by the MEME web server. Motif distribution includes different colored boxes, each representing a unique numbered motif as indicated in the legend. The width differences among the boxes represent the motif length. (B) Exon/intron structures of MYB genes from C. moschata. TBtools software was used to visualize the above results. The exons and introns are presented as filled blue sticks and thin gray single lines, respectively. Upstream and downstream regions are represented by black bars at the two ends of sequences.

Employing the MEME tool facilitated the identification of conserved motifs, thus enhancing the comprehension of CmoMYB protein diversity within the pumpkin genome. The findings, illustrated in Table 1, delineated the presence of 10 conserved motifs. Fig. 2 depicted that numerous CmoMYBs predominantly comprised motifs 1, 2, 3, 4, 5, 6, and 7/8. While motifs typically occur singularly, exceptions were noted; for instance, motif 3 was repeated in CmoMYB115 and CmoMYB173 but was absent in CmoMYB151. Similarly, motif 4 exhibited duplication in CmoMYB30 but was not detected in CmoMYB159. Moreover, closely related CmoMYB proteins tended to exhibit analogous motifs, suggesting functional similarities within specific subgroups. Notably, certain CmoMYBs displayed unique motifs; for instance, motif 9 exclusively appeared in the C12 and C13 subfamilies. This underscores the potential involvement of specific motifs in executing distinct functions within MYB proteins.

Table 1 Specific conserved motifs identified by MEME among CmoMYB proteins in C. moschata.

Motif	Width	Sites	Logo	Consensus sequence	e-value	
1	21	171		RCGKSCRLRWINYLRPDJKRG	2.2e-2900	
2	21	167		RWSKIAAQLPGRTDNEIKNYW	1.2e-2785	
3	21	171		KKGPWTPEEDEKLINYIQKHG	3.8e-2013	
4	15	171		EEEELIIELHALLGN	1.2e-1129	
5	11	124		NWRSLPKNAGL	9.8e-566	
6	11	85		MGRAPCCDKAG	6.6e-553	
7	27	42		NTHJKKKLJKMGIDPVTHKPISDLLDL	1.2e-539	
8	8	82		NTHLKKKL	1.2e-160	
9	40	9		RTRVQKQAKQLKCDVNSKQFKDTMRYLWIPRLVERIQASS	3.9e-133	
10	11	41		PRNWSLIAESJ	7.5e-117	

Cis-acting element prediction of CmoMYB gene promoters

To explore the transcriptional regulatory properties of CmoMYB genes, cis-acting elements were predicted by using PlantCARE online software (Fig. 3). The analysis unveiled a pervasive distribution of stress- and hormone-responsive elements within the promoters of CmoMYB genes, in addition to numerous core cis-elements. Enumeration of distinct cis-elements revealed that abscisic acid-responsive elements (ABRE, 557) were the most prevalent in the CmoMYBs’ promoters, succeeded by MeJA-responsive elements (TGACG-motif and CGTCA-motif, 547) and salicylic acid-responsive elements (SARE and TCA-element, 124). Furthermore, these promoters encompassed auxin-responsive elements (TGA, AUXRE), drought-induced response elements (MBS), flavonoid biosynthetic regulation (MBSI), low temperature-responsive elements (LTR), defense and stress-response elements (TC-rich repeats), and various other cis-acting elements.

Figure 3 Cis-acting elements in the promoter region of ComMYB genes.

The cis-acting elements were identified by PlantCARE and visualized using the TBtools software. Different colors of the box indicate different cis-acting elements.

Expression analysis of the CmoMYB genes

Utilizing expression data extracted from the Cucurbitaceae genome database, a heat map delineating the expression patterns of CmoMYB genes across four distinct tissues was generated using TBtools (Fig. S3). Approximately 90 of these 175 (51.4%) CmoMYBs showed the highest expression level in root, 59 (34%) in stem, 37 (21%) in leaf, and 34 (19%) in fruit. Besides, a total of 13 CmoMYB genes from different subgroups (Fig. 1) were selected for expression analysis by qRT-PCR. The expression levels of 13 CmoMYBs were performed in three tissues (root, stem, and leaf) at the three-leaf seedlings stage (Fig. 4). A variety of expressions of these CmoMYBs were found in the three tissues. Most CmoMYBs including CmoMYB99, CmoMYB142, CmoMYB154, CmoMYB144, CmoMYB116, CmoMYB70, CmoMYB46, CmoMYB3, and CmoMYB9 were significantly up-regulated in the roots, several CmoMYB genes including CmoMYB165, CmoMYB142, CmoMYB59, and CmoMYB29 were highly expressed in the leaves, while CmoMYB64 and CmoMYB59 genes were highly expressed in the stem, indicating that these CmoMYBs might be involved in the various biological processes in the different tissues.

Figure 4 Expression levels of the selected 13 CmoMYB genes in different tissues.

Expression levels of the selected 13 CmoMYB genes (CmoMYB99, 165, 142, 154, 144, 116, 70, 46, 64, 59, 3, 29, and 9) in different tissues (Student’s t-test; *p < 0.05, ***p < 0.001).

The expression levels of CmoMYB genes under hormones and abiotic stresses

The expression levels of many CmoMYB genes were significantly changed after ABA treatment (Fig. 5). ABA treatment for 3 h induced the expression levels of 10 CmoMYBs at a degree from 1.23-fold to 65.33-fold, while CmoMYB99 was not affected, and reduced the expression of CmoMYB165 and CmoMYB154 to 77.3% and 63.9%, respectively. The CmoMYB genes were significantly regulated after 6 h after ABA treatment (10 up-regulated and 1 down-regulated), except for CmoMYB64 and CmoMYB59, which returned to initial values. As shown in Fig. 6, 11 CmoMYB genes were significantly regulated at 3 h under JA treatment (9 up-regulated and 2 down-regulated), except for CmoMYB59 (which started to decrease at 6 h) and CmoMYB3 (which started to increase at 6 h). The transcription levels of 7 CmoMYB genes were less significant at 6 h than at 3 h, such as, the expression levels of CmoMYB99 decreased from 50.70-fold to 11.93-fold, CmoMYB116 decreased from 15.94-fold to 7.79-fold, and even CmoMYB142 and CmoMYB29 did not change significantly at 6 h. After SA treatment, all CmoMYB genes were significantly regulated at 3 h (11 up-regulated and two down-regulated) (Fig. 7). After 6 h of treatment, the expression level of 12 CmoMYB genes started to decrease, except for CmoMYB144, which was up-regulated. Compared with 0 h, eight CmoMYB genes had no significant change, only three CmoMYB genes had a significant change, and two CmoMYB genes were still continuously down-regulated at 6 h. Drought treatment for 6 h significantly increased the transcript levels of 10 CmoMYB genes (CmoMYB99, 165, 142, 144, 116, 70, 46, 59, 3, and 9) with distributions increased by 43.01-, 1.55-, 4.32-, 12.44-, 5.78-, 12.72-, 4.96-, 4.18-, 9.02-, and 9.43-fold, respectively, while CmoMYB29 decreased to 77%, and CmoMYB64 and CmoMYB154 did not change significantly (Fig. 8). Interestingly, seven genes (CmoMYB165, 142, 154, 70, 46, 59, and 9) appeared to be down-regulated at 3 h and significantly up-regulated at 6 h.

Figure 5 ABA-induced expression patterns of 13 CmoMYB genes.

The uniformly sized three-leaf pumpkin seedlings were treated with 10 µM ABA. The leaves were harvested at the indicated times for RNA extraction and qRT-PCR analysis. The Cmoβ-actin gene was used as the normalization reference gene. The relative expression level of genes was calculated by the 2−∆∆CT method. Three experimental replicates were performed for each sample (Student’s t-test; *p < 0.05, **p < 0.01, ***p < 0.001).

Figure 6 The expression levels of 13 CmoMYB genes under JA (100 µM).

The three-leaf pumpkin seedlings leaves were harvested at the indicated times for RNA extraction and qRT-PCR analysis. The Cmoβ-actin gene was used as the normalization reference gene. The relative expression level of genes was calculated by the 2−∆∆CT method. Three experimental replicates were performed for each sample (Student’s t-test; *p < 0.05, **p < 0.01, ***p < 0.001).

Figure 7 SA-induced expression patterns of 13 CmoMYB genes.

The three-leaf pumpkin seedlings’ leaves were harvested at the indicated times after being treated with 100 µM SA for RNA extraction and qRT-PCR analysis. The Cmoβ-actin gene was used as the normalization reference gene. Three experimental replicates were performed for each sample (Student’s t-test; **p < 0.01, ***p < 0.001).

Figure 8 The expression levels of 13 CmoMYB genes under simulated drought (20% PEG6000).

The leaves were harvested at the indicated times for RNA extraction and qRT-PCR analysis. The Cmoβ-actin gene was used as the normalization reference gene. Three experimental replicates were performed for each sample (Student’s t-test; *p < 0.05, ***p < 0.001).

Discussion

The MYB gene family has been systematically characterized across diverse plant species, including Chinese pear, rice, Arabidopsis, and soybean (Du et al., 2012; Dubos et al., 2010; Cao et al., 2016; Dai et al., 2012). Despite extensive investigations in these species, the CmoMYB gene family in Cucurbita moschata remains relatively understudied, with its functional roles yet to be elucidated. This current investigation represents the foundational analysis of the MYB superfamily genes within C. moschata, leveraging the available genomic resources as detailed by Sun et al. (2017). A comprehensive survey revealed 175 proteins harboring MYB repeats. The Mw and pI of these proteins emerged as crucial parameters influencing their molecular and biochemical functionalities (Xia et al., 2017). Notably, our analysis unveiled substantial variations in both size and pI among CmoMYB proteins, a trend reminiscent of findings reported by Li et al. (2020), implying potential context-dependent functional diversity among CmoMYB proteins. Typical TFs encompass four pivotal domains, including nuclear localization signal, DNA-binding, transcriptional regulatory sites, and oligomerization domains (Du et al., 2012). Protein subcellular localization predictions show that all CmoMYB proteins are localized to the nucleus. The chromosomal distribution analysis revealed that CmoMYBs are dispersed across the 20 chromosomes of C. moschata. however, this distribution appears uneven. Chromosome 14 emerged as the locus harboring the highest number of CmoMYB genes, followed by chromosome 1, while chromosome 16 exhibited the fewest CmoMYB genes. Furthermore, numerous CmoMYBs tend to cluster in distinct genomic regions, displaying notably elevated densities towards the chromosomal telomeres (Fig. S1). This pattern of ComMYB gene distribution bears resemblance to observations documented in prior studies across other species, such as chili peppers (Arce-Rodríguez, Martínez & Ochoa-Alejo, 2021) and potatoes (Li et al., 2020).

As part of this study, a phylogenetic tree was constructed for CmoMYBs and AtMYBs in order to further investigate their orthologous relationship (Fig. 1). The resulting phylogenetic tree revealed 33 distinct categories. Our findings align with previous research, indicating that members within the same branch share conserved functions, likely stemming from a common ancestor (Kranz et al., 1998; Stracke, Werber & Weisshaar, 2001). Most of the CmoMYB proteins were clustered with AtMYB proteins. For example, the CmoMYBs of CmoMYB104, CmoMYB38, CmoMYB36, CmoMYB40, CmoMYB30, CmoMYB102, and CmoMYB26 grouped with AT3G62610, AT2G47460, and AT5G49330 to form clade C26 (The syntenic analysis between these proteins was shown in Fig. S4), which may indicated that these CmoMYB genes might related to the biosynthesis of flavonoids (Stracke et al., 2007; Ballester et al., 2009; Stracke et al., 2010); clade C11 was consists of CmoMYB83, CmoMYB94, CmoMYB86, and CmoMYB96 of pumpkin and the reported proteins AT3G27810, AT5G40350, AT3G01530 (the syntenic analysis between these proteins was shown in Fig. S5), which are involved in the control of PAL genes and the elongation of staminal filaments (Sablowski et al., 1994; Cheng et al., 2009). Conversely, clades C16 and C24 lacked AtMYBs, indicating that certain CmoMYB genes may be unique to pumpkin. Overall, the clustering patterns offer valuable insights into the roles of CmoMYB proteins. Moreover, the analysis of gene structure and motif protein alignment strongly supports subgroup classification (Fig. 2). Consistent with prior studies, our observations indicate that MYB genes within the same subgroup typically exhibit similar exon-intron structures, highlighting their high conservation across species (Du et al., 2012; Jiang & Rao, 2020).

MYB proteins feature a structurally dynamic region, responsible for regulatory activities, alongside a conserved MYB structural domain, which facilitates the recognition of target gene promoters (Ambawat et al., 2013). Analysis of the motif results showed that CmoMYB proteins belonging to the same subgroup shared the common motifs, suggesting that they might have similar functions. In addition, we analyzed the expression profile of CmoMYB genes in different tissues (Fig. S3). 90 CmoMYBs had the highest level of transcript accumulation in root tissue, 59 CmoMYBs in stem tissue, 37 CmoMYBs in leaf tissue, and 34 CmoMYBs in seed tissue. In the analysis of 13 genes, most CmoMYBs in roots were significantly up-regulated, and CmoMYB29 was expressed only in leaves. In contrast, CmoMYB64 and CmoMYB59 genes were highly expressed in stems, suggesting that these CmoMYBs may be involved in various biological processes in different tissues, such as root growth, stem elongation, and leaf development (Fig. 4).

Cis-acting elements act as sites for specific binding of transcription factors and are involved in the regulation of gene expression (Zhang, Wu & Yue, 2020). To explore whether the MYB gene responds to adversity stress through cis-acting elements, we delineated the cis-acting elements in the promoter of the CmoMYBs (Fig. 3). Prominently, abscisic acid-responsive elements predominated, succeeded by MeJA-responsive elements, salicylic acid-responsive elements, drought-induced response elements, as well as defense and stress-response elements, among others. The findings imply the potential involvement of the CmoMYB gene family in abiotic stress and hormone-mediated responses. Moreover, the differential transcriptional regulation observed among various CmoMYB gene types underscores the diverse functional roles played by CmoMYBs in cellular processes. Guided by these predictions, a subset of 13 CmoMYB genes was scrutinized for their responsiveness to three distinct hormones and simulated drought stress conditions. Differential stress responses among these genes were observed, potentially attributed to the distinct repertoire of cis-acting elements harbored within their promoters, with notable regulation evident across most CmoMYB genes (Figs. 3 and 5–8). Furthermore, our inquiry into the responsiveness of these genes to various stressors revealed a spectrum of regulatory patterns, possibly linked to the diversity and abundance of cis-acting elements present in their promoters. Noteworthy prior research highlights the significance of OsMYB1R1, TaMYB31, GaMYB85, and SsMYB113 in mediating drought stress responses (Peng et al., 2023; Zhao et al., 2018; Butt et al., 2017; Zhang et al., 2022). In our investigation, CmoMYB99, CmoMYB144, CmoMYB116, CmoMYB70, and CmoMYB3 exhibited heightened expression under drought stress, contrasting with the down-regulation of CmoMYB29. These findings underscore the nuanced transcriptional dynamics under varying stress contexts, indicative of intricate responses mediated by complex signaling pathways (Xia et al., 2017; Yang et al., 2016; Yang et al., 2018; Gill et al., 2016). Collectively, our results underscore the pivotal role played by the CmoMYB gene family in orchestrating responses to diverse stress stimuli.

Conclusions

The MYB superfamily, ubiquitously distributed across eukaryotes, assumes pivotal roles in governing plant growth and development. Notably, environmental stresses, exemplified by drought, constitute significant natural threats to crop plants. The discernment of genes responsive to abiotic stress emerges as a potentially efficacious strategy for augmenting resistance to such adversities in pumpkins. Within the purview of this investigation, a comprehensive analysis of the entire genome identified 175 MYB proteins. While these proteins exhibited a high degree of conservation in motif composition, their expression profiles varied across different tissues, indicative of the nuanced activity and diversity inherent in CmoMYBs, governing pumpkin growth. To elucidate the functional attributes of CmoMYBs, the prediction of cis-acting elements for these genes revealed the presence of abiotic and phytohormone-responsive elements within their promoters. This observation suggests a putative role for these genes not only in the regulation of plant growth and development but also in mounting responses to adverse conditions. Delving further into the functional characterization of CmoMYB proteins, we scrutinized the expression patterns of 13 CmoMYB genes across distinct branches of the phylogenetic tree under various abiotic stresses, including ABA, SA, MeJA, and drought. The outcomes demonstrated universal responsiveness of all CmoMYB genes to these treatments, with discernible variations in expression levels under distinct stress conditions. These findings furnish a foundational framework for prospective inquiries aimed at delineating the functional attributes of the MYB gene family within the context of pumpkin physiology.

Supplemental Information

Supplemental Information 1 The Information of RT-qPCR Experiments (MIQE).

Supplemental Information 2 RT-qPCR data.

Supplemental Information 3 The information of the identified CmoMYB genes.

Supplemental Information 4 The specific primers for the ComMYB genes for qRT-PCR analysis.

Supplemental Information 5 The homologous MYB gene pairs were identified in C.moschata.

Supplemental Information 6 The chromosomal distribution of CmoMYB genes.

The chromosomal position of each CmoMYB was mapped according to the C.moschata genome. Gff3 files was downloaded from the cucurbitaceae database, TBtools software was used to visualize the result. The numbers of chromosomes are indicated in the left middle of each chromosome, the right side of the chromosome is the location of the gene. Scale bar is in million bases (Mb).

Supplemental Information 7 Chromosomal localization of CmoMYB genes.

The chromosomal position of each CmoMYB was mapped according to the C.moschata genome. The different colorful lines in circle indicate a collinearity relationship among genes.

Supplemental Information 8 Expression profiles of ComMYB genes in different tissues (leaves, roots, stem, and fruit).

Higher and lower levels of transcript accumulation are indicated by red and blue, respectively. The expression data were obtained from the cucurbitaceae database, and the heat map for specific expression analysis was constructed by TBtools software.

Supplemental Information 9 The multiple sequence alignments between CmoMYB and AtMYB proteins.

Multiple sequence comparison of the amino acid sequences of CmoMYB104, CmoMYB38, CmoMYB36, CmoMYB40, CmoMYB30, CmoMYB102, CmoMYB26, AT3G62610, AT2G47460, and AT5G49330. Firstly, the MEGA software was used to do multiple sequence comparison of amino acid sequences. Then, visualization using BEG (BEGinner) mode in ESPript 3.0 online website (https://espript.ibcp.fr/ESPript/cgi-bin/ESPript.cgi)

Supplemental Information 10 The multiple sequence alignments between CmoMYBs and AtMYBs.

Multiple sequence comparison of the amino acid sequences of CmoMYB83, CmoMYB94, CmoMYB86, CmoMYB96, AT3G27810, AT5G40350, and AT3G01530. Firstly, the MEGA software was used to do multiple sequence comparison of amino acid sequences. Then, visualization using BEG (BEGinner) mode in ESPript 3.0 online website (https://espript.ibcp.fr/ESPript/cgi-bin/ESPript.cgi)

Additional Information and Declarations

Competing Interests

Author Contributions

Data Availability

The authors declare that they have no competing interests.

Minyan Xu conceived and designed the experiments, performed the experiments, analyzed the data, prepared figures and/or tables, authored or reviewed drafts of the article, and approved the final draft.

Jingjing Fu conceived and designed the experiments, analyzed the data, prepared figures and/or tables, and approved the final draft.

Ying Ni performed the experiments, authored or reviewed drafts of the article, and approved the final draft.

Chenchen Zhang conceived and designed the experiments, analyzed the data, authored or reviewed drafts of the article, and approved the final draft.

The following information was supplied regarding data availability:

The raw measurements are available in the Supplemental Files.

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
