# Peer review of "Genome‑wide analysis of the MYB gene family in pumpkin"

_PeerJ, doi:10.7717/peerj.17304_

## Round 0.1 · original submission · Major Revisions

I am writing to inform you that we have received and carefully reviewed feedback from two independent referees regarding your manuscript. While your work significantly contributes to our understanding of the MYB gene family in pumpkin, I have reached the conclusion that it would be premature to publish your paper in its present form. I recommend that you consider revising the manuscript to address the major concerns raised by the referees.

Referee #1 has recommended several areas for improvement. It is crucial to place your work in a better context, particularly in the introduction and discussion sections, by emphasizing the relevance of studying the regulation of abiotic stress in Cucurbita moschata. Additionally, the referees suggest using HMM (Hidden Markov Models) for more accurate identification of MYB members. I recommend employing the MYB-annotator software (https://doi.org/10.1186/s12864-022-08452-5), specifically designed for identifying MYB transcription factors in plants. Furthermore, authors should perform statistical analyses to determine differential gene expression for different treatments, tissues, and time points. Other major recommendations concern the need to clarify the methods used and to enhance the quality of the figures. Several additional suggestions were made regarding data analysis and result presentation.

To assist you in the revision process, I recommend that you provide a detailed response to each of the referees' comments and outline the specific changes you intend to make in response to their feedback. Please ensure that you also provide a revised version of the manuscript with all modifications clearly marked.

We value your contributions to PeerJ and look forward to receiving your revised manuscript. If you have any questions or require further clarification regarding the referees' comments or the revision process, please do not hesitate to contact me. We are committed to working with you to ensure the successful resubmission of your work.

Sincerely,
Francisco Balao

Reviewer 1 ·

Basic reporting

Xu et al. performed genome-wide identification and analysis of the MYB gene family in pumpkin. Their results implicated that MYB genes play important roles in hormone responses
and abiotic stresses.

The main concerns list as follows:

1. Fig 1 and Fig 2 are not clear, and I can hardly make out the details.

2. You should perform t-test for Fig 4, 5, 6, 7 and 8.

3. The contents of line 41-43 are confusing.

4. In this manuscript, your primary focus has been on the regulation of abiotic stress by CmoMYBs. Therefore, it is important to provide an explanation in the introduction or discussion as to why this focus is significant. For instance, you could address whether Cucurbita moschata frequently faces drought stress, thereby emphasizing the relevance of studying abiotic stress responses in these genes.

5. You have provided two types of expression data for CmoMYBs in different tissues. It would be beneficial to compare the data for consistency in the results.

6. The contents of line 298-302 are confusing. Please check it and make revisions.

I found a few minor errors:

1. In line 189-190, please delete one “related”.
2. In line 230, “CmoMYB164” should change into “CmoMYB64”.
3. In line 233, “20” should change into “2”.
4. You should indicate to the readers where the data of subcellular localization can be found, Table S1, right?
5. The legend of Fig.S1 has some errors. “The numbers of chromosomes are indicated in the right middle of each chromosome, the left side of the chromosome is the location of the gene.” should change into “The numbers of chromosomes are indicated in the left middle of each chromosome, the right side of the chromosome is the location of the gene.”.

Experimental design

7. In line 79-82, you would better use HMMER for searching MYB members, rather than solely conducting a blast comparison between Arabidopsis and pumpkin.

8. You should explain the criteria you used for naming your genes, such as homology with Arabidopsis or their chromosomal locations?

9. You should provide the method you used to create the gene location map. Additionally, you should provide the methods/tools you used for visualizing the exon-intron structure.

10. After PEG treatment, did you measure any physiological data, such as whether the plants exhibited signs of water deficiency? Beginning with the phenotype and subsequently delving into the underlying molecular mechanisms would enhance the overall logical flow and persuasiveness of the experiment.

11. For lines 271-286, to establish more comprehensive orthologous relationships between CmoMYBs and AtMYBs, it would be advisable to conduct a syntenic analysis to enhance the persuasiveness of your results or inferences.

Validity of the findings

12. In abstract, the last two sentences of the abstract are somewhat confusing. Expression analysis in different tissues does not allow you to conclude that CmoMYBs have divergent functions in regulating abiotic/biotic stresses. Additionally, the data presented in this manuscript can only suggest some implications of CmoMYBs' function in abiotic stress (specifically drought), and there isn't enough evidence to demonstrate that CmoMYBs play significant roles in biotic stresses.

13. In line 72-74, the objectives of this study are somewhat overstated.

14. In lines 322-324, how did you arrive at the conclusion that “In this study, the CmoMYB85 / 113 / 125 / 43 / 56 proteins are very close to these CsMYB proteins, suggesting that they may be involved in plant-pathogen interactions.”?

15. The conclusion appears to be disorganized. Instead of solely concentrating on biotic stress, you should consider rewriting it for clarity.

Reviewer 2 ·

Basic reporting

(1)Clear and unambiguous, professional English used throughout.
(2)Literature references, sufficient field background/context provided.
(3)Professional article structure, figures, tables. Raw data shared.
(4)Self-contained with relevant results to hypotheses.

Experimental design

(1)Original primary research within Aims and Scope of the journal.
(2)Research question well defined, relevant & meaningful. It is stated how research fills an identified knowledge gap
(3)Rigorous investigation performed to a high technical & ethical standard.

Validity of the findings

(1)Impact and novelty not assessed. Meaningful replication encouraged where rationale & benefit to literature is clearly stated.
(2)All underlying data have been provided; they are robust, statistically sound, & controlled.
(3)Conclusions are well stated, linked to original research question & limited to supporting results.

Additional comments

This study was the first to identify the pumpkin MYB gene family and analyze its expression under abiotic stress.However, the following related issues need to be addressed。
1. MYB transcription factors (TFs) including 1R-MYB, R2R3-MYB, 3R-MYB and 4R-MYB. So, how many these types MYBs in pumkin?
2. Abbreviations should be written for the first time in the article, for example Line 91 CDS.
3. Variety requires single quotes, for example Line 115 "TianMiyihao"
4. The gene related to tissue expression profile needs to be verified by qRT-PCR。

---

## Round 0.2 · Major Revisions

The manuscript has been thoroughly reviewed by three referees, and I apologize for the delay in providing this feedback. The new version of the manuscript shows notable improvements. However, further revisions are necessary to enhance its quality. The manuscript requires significant revisions, primarily focusing on the phylogenetic analysis where a shift from the neighbor-joining to the maximum likelihood method is advised for more accurate results. A detailed expansion on functional predictions of MYB genes, akin to studies on Arabidopsis thaliana, is necessary. Additionally, the inclusion of gene duplication and collinearity analysis within the pumpkin MYB gene family is recommended. The manuscript should also specify data sources (public database accession numbers) and provide a comprehensive description of the RNA sequencing methodologies used. Clarification is needed on the selection criteria for the 13 CmoMYB genes chosen for qRT-PCR analysis. Figure 4 should include statistical analyses , and the Discussion section needs reorganization to focus more sharply on the specific scientific questions and problems addressed by the research. The current introduction of the manuscript is overly descriptive and tends to repetitively preview the results. Finally, the manuscript should include a description of genes known to respond to abiotic stress. In addition, the referee#3's recommendation indicates a need for enhancement in the English language quality of the manuscript.

**Language Note:** The Academic Editor has identified that the English language must be improved. PeerJ can provide language editing services - please contact us at [email protected] for pricing (be sure to provide your manuscript number and title). Alternatively, you should make your own arrangements to improve the language quality and provide details in your response letter. – PeerJ Staff

Reviewer 1 ·

Basic reporting

In most cases, I did not find the revisions in the final pdf documents, although the authors made the corresponding responses. Moreover, according to authors' instructions such line(XX-XX), I did not find the corresponding revisions in the manuscript. Please re-revise and make sure to let reviewer review clearly and easily.

Experimental design

NA

Validity of the findings

NA

Additional comments

NA

Reviewer 2 ·

Basic reporting

(1)Clear and unambiguous, professional English used throughout.
(2)Literature references, sufficient field background/context provided.
(3)Professional article structure, figures, tables. Raw data shared.

Experimental design

(1)Original primary research within Aims and Scope of the journal.
(2)Rigorous investigation performed to a high technical & ethical standard.
(3)Methods described with sufficient detail & information to replicate.

Validity of the findings

not comment

Reviewer 3 ·

Basic reporting

In this manuscript, authors identified MYB genes from Cucurbita moschata and analyzed their classification and structure. Also, the expression level of 175 MYB genes was displayed and used the qRT-PCR to further verify the response for exogenous phytohormone exposure and abiotic stress conditions. This is very meaningful for further studies to explore the role of the MYB gene family under abiotic stresses in pumpkin. However, the English writing should be further improved. The detailed critical issues of the manuscript is as follows:

1.Xu et al have shown that phylogenetic tree was constructed using the neighbor-joining method in Lines 156 is not the most appropriate for this analysis. A tree based on maximum likelihood method should be build to further verify this result.
2.As described by the author in line 188 “those from the same subgroup likely have similar functions.”, authors should provide a more detailed description of the functional prediction in Line 156-162. As we known, the MYB gene function of Arabidopsis thaliana has been thoroughly investigated.
3.Line 163-173, in Results “3.2”, I suggest that the authors add analysis of duplication events and collinearity of MYB family in pumpkin.
4.Line 206, authors should provide data sources (NCBI, NO. of project) and citations. In Methods, the detailed RNA sequencing methods should be described.
5.Line 208-209, “13 CmoMYB genes from different subgroups were chosen for expression analysis using qRT-PCR.”, author should explain why these 13 genes were chosen.
6.Line 214-216, difference significance analysis should be added in Fig. 4
7.The Discussion section should be reorganized. Too much description of the results is unnecessary. Authors should discuss questions that are closely related to the results of the research, that is, the more important scientific questions in the small field. The purpose of the discussion is to clarify what specific problems in the field your research solves.
8.Line 69-70, some reported genes responding to abiotic stress should be described.

Experimental design

no comment

Validity of the findings

no comment

Additional comments

no comment

---

## Round 0.3 · Minor Revisions

Following the insightful feedback from the referees and my own review, it's clear that the manuscript has significantly improved since its initial submission. The authors have diligently addressed many of the concerns previously raised, enhancing both the clarity and depth of the study. However, there are still a few minor changes necessary to ensure the manuscript meets the highest standards of publication. Below, I have outlined some editorial comments that should be considered to refine the manuscript further. These adjustments are aimed at clarifying certain sections, improving consistency in formatting and terminology, and ensuring that all figures accurately represent the data. After incorporating these minor adjustments, I am confident that the manuscript will be fully prepared for publication.

Scientific Names and Italicization:
◦ Line 79: Change to "Arabidopsis thaliana and Oryza sativa".
◦ Line 8: Put "Oryza sativa" in italics.
2. Text Modifications:
◦ Line 139: Change to "For tissue-specific expression analysis of 175 CmoMYB genes, the expression data for four tissues (leaves, roots, stem, and fruit) were obtained from the Cucurbitaceae database."
◦ Lines 148-152: Revise to "A total of 175 CmoMYB proteins were successfully identified (see Table S1 for details), including 1R-MYB (21), R2R3-MYB (149), 3R-MYB (4), and 4R-MYB (1)."
◦ Line 157: Correct to “were”.
◦ Line 164: Change to “circles in Figure 1)”.
◦ Line 171: Start with “The 175 CmoMYB genes were mapped to…”
◦ Lines 181-185: Clarify the confusing paragraph. Possibly rephrase to explain if "142 out of 175 MYB genes were duplicated" and why only "135 were located on chromosomes". This needs better explanation in the methods section.
◦ Lines 187-188: Revise to "analyzed for 175 CmMYBs (Figure 2).”
3. Section and Content Reorganization:
◦ Lines 217-220: This sentence should be relocated to the Discussion section.
◦ Line 227: Eliminate “Primer design...”. It is redundant.
◦ Line 237: Remove the sentence “The expression of…”.
◦ Line 239: Amend to “significantly changed after ABA treatment”.
◦ Line 243: Revise to “which returned to initial values.”
◦ Lines 244-245: Reposition “under JA treatment” to follow “regulated at 3h”.
4. Figures:
◦ Figure 3: Ensure the significance asterisk is shown.
◦ Figure 8: Question for the authors: Why not follow the same style as previous barplot figures?
5. Word Choice and Removal of Details:
◦ Line 265: Replace “inaugural” with “foundational”.
◦ Line 274: Omit mention of the software.
◦ Line 276: Revise to “C. moschata. However…”
◦ Lines 301-303: Delete the sentence. It is duplicated.
◦ Lines 315-317: Authors should contextualize the findings rather than merely repeating results. Include additional references.

Reviewer 1 ·

Basic reporting

The manuscript has undergone extensive revision. However, I have two comments to address:
1. For Fig S4 and S5, at most they can be referred to as multiple sequence alignments rather than syntenic analysis.
2. Please provide the methods used to identify the duplicated MYB gene pairs.

Experimental design

NA

Validity of the findings

NA

Additional comments

NA

Reviewer 2 ·

Basic reporting

no comment

Experimental design

no comment

Validity of the findings

no comment

Reviewer 3 ·

Basic reporting

The author has fully resolved the problems I raised, so I proposes to accept this manuscript.

Experimental design

None

Validity of the findings

None

Additional comments

None

---

## Round 0.4 · Minor Revisions

I hope this message finds you well. We kindly ask you to revise Figure 4. Ensure the significance asterisk is shown.

Could you please submit the revised figure at your earliest convenience? If you have any questions or need further clarification, please feel free to reach out.

---

## Round 0.5 · accepted · Accept

I am pleased to inform you that your manuscript has been accepted for publication.